# GPEN: Global Position Encoding Network for Enhanced Subgraph Representation Learning

Nannan Wu [* 1]   Yuming Huang [* 1]   Yiming Zhao [1]   Jie Chen [2]   Wenjun Wang [1 3]

## Abstract

Subgraph representation learning has attracted growing interest due to its wide applications in various domains. However, existing methods primarily focus on local neighborhood structures while overlooking the significant impact of global structural information, in particular the influence of multi-hop neighbors beyond immediate neighborhoods. This presents two key challenges: how to effectively capture the structural relationships between distant nodes, and how to prevent excessive aggregation of global structural information from weakening the discriminative ability of subgraph representations. To address these challenges, we propose GPEN (Global Position Encoding Network). GPEN leverages a hierarchical tree structure to encode each node's global position based on its path distance to the root node, enabling a systematic way to capture relationships between distant nodes. Furthermore, we introduce a boundary-aware convolution module that selectively integrates global structural information while maintaining the unique structural patterns of each subgraph. Extensive experiments on eight public datasets identify that GPEN significantly outperforms state-of-the-art methods in subgraph representation learning.

## 1. Introduction

Graphs are widely used to model relational data, such as transportation networks (Jiang & Luo, 2022), biological systems (Fout et al., 2017), and social networks (Chen et al., 2018). In recent years, Graph Neural Networks (GNNs)

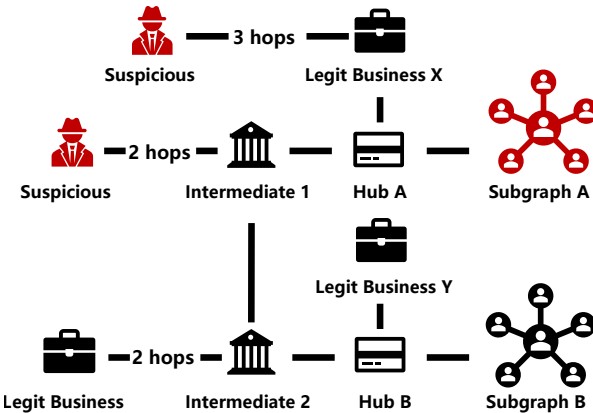

Figure 1. Illustration of the challenge. While fraudulent (Subgraph A) and legitimate (Subgraph B) transaction schemes exhibit similar local structures centered around hub nodes and regular transaction patterns, Subgraph A receives funds from suspicious sources through multiple hops of intermediate accounts, highlighting the importance of global structural information beyond neighborhoods.

have achieved remarkable success in graph-related tasks, including node classification (Chen et al., 2020; Xiao et al., 2022), link prediction (Lei et al., 2019; Long et al., 2022), and graph classification (Zhang et al., 2018; Xie et al., 2022; Wei et al., 2023).

Subgraph representation learning, which focuses on capturing meaningful representations of subgraphs within a larger graph, has attracted growing interest. It has shown strong potential in various applications, such as predicting cellular functions in protein-protein interaction networks and diagnosing rare diseases in phenotype knowledge graphs (Sharan et al., 2007; Alsentzer et al., 2020b).

However, existing methods focus mainly on local neighborhood structures while overlooking the significant impact of global structural information (Alsentzer et al., 2020b; Wang & Zhang, 2022; Jacob et al., 2023; Kim & Oh, 2024), in particular the influence of multi-hop neighbors beyond immediate neighborhoods. As illustrated in Figure 1, in financial transactions, money laundering schemes often deliberately mimic legitimate transaction patterns to avoid detection. Both fraudulent (Subgraph A) and legitimate (Subgraph B) schemes appear similar when only examining

*Equal contribution [1]College of Intelligence and Computing, Tianjin University, Tianjin, China [2]College of Management and Economics, Tianjin University, Tianjin, China [3]Yazhou Bay Innovation Institute, Hainan Tropical Ocean University, China. Correspondence to: Nannan Wu <nannan.wu@tju.edu.cn>.

*Proceedings of the 42nd International Conference on Machine Learning*, Vancouver, Canada. PMLR 267, 2025. Copyright 2025 by the author(s).

their direct transaction behaviors, as they both contain a central hub node that processes multiple regular transactions with surrounding accounts. This structural similarity arises because fraudsters intentionally construct their schemes to resemble normal business operations. The key distinction is the global structural information: fraudulent schemes (Subgraph A) receive funds from suspicious sources through multiple hops of intermediate accounts, deliberately creating distance between the suspicious source and the final hub node. This indicates that capturing only local neighborhood structures is inadequate for learning effective subgraph representations.

Modeling the global structural information presents two major challenges. First, effectively capturing the structural relationships between distant nodes is challenging. Current methods, such as random walk sampling, may overlook important distant nodes and struggle to encode sampled sequences into meaningful global features. This requires capturing global structural information from a more unified perspective rather than relying on fragmented local views. Moreover, simply aggregating global structural information can weaken the discriminative ability of subgraph representations. According to the homophily principle, nodes connected by edges are more likely to share similar properties. Excessive aggregation of global structural information may introduce distant nodes that are irrelevant to the subgraph, weakening the learning of its intrinsic structure. This highlights the need for a module that can selectively integrate global structural information while preserving the inherent structural characteristics of the subgraph.

To address these challenges, we propose GPEN (Global Position Encoding Network), which effectively captures global structural information while preserving inherent subgraph structures. GPEN leverages a hierarchical tree structure to encode global position information and control information aggregation through boundary-aware convolution. Specifically, GPEN first transforms the graph into a tree structure to utilize its natural hierarchical organization for position encoding. Using this tree structure, we encode each node's global position based on its path distance to the root node, enabling a structured way to capture relationships between distant nodes. This tree-based position encoding effectively captures multi-hop relationships and addresses the challenge of modeling global structural information systematically, rather than relying on fragmentary local views from sampling. Furthermore, as an optional enhancement, we explore additional benefits of the tree structure and discover a tree perturbation trick that utilizes the tree's hierarchical nature to generate consistently perturbed samples, which helps stabilize model training. To prevent excessive aggregation of global structural information that could obscure intrinsic subgraph structures, we introduce a boundary-aware convolution module. This module computes difference vectors

between nodes to control information flow during convolution, allowing the model to selectively integrate global structural information while maintaining the unique structural patterns of each subgraph. It helps maintain the subgraph's inherent structural characteristics while incorporating useful information from the global structural information.

We conclude the contributions of GPEN as follows:

- We propose a novel tree-based global position encoding that provides a unified way to capture multi-hop relationships for all nodes in the graph, effectively addressing the challenge of modeling global structure.

- We design a boundary-aware convolution module that selectively integrates global structural information while preserving intrinsic subgraph structures, preventing the dilution of structural characteristics.

- Extensive experiments on eight public datasets identify that GPEN outperforms state-of-the-art methods, achieving superior performance in subgraph representation learning.

## 2. Preliminaries

**Notations.** Let $G = (V, E)$ denote a undirected graph with node set $V$ and edge set $E \subseteq V \times V$. $S = \{S_1, S_2, ..., S_k\}$ is a set of subgraph of $G$. For each subgraph $S_i = (V_i, E_i)$ has a label $y_i$ , where $V_i \subseteq V$ and $E_i \subseteq E$. Furthermore, the subgraphs may share nodes and/or edges and consist of multiple components.

**Problem Definition.** Given a set of subgraphs $S = \{S_1, S_2, ..., S_k\}$ in a graph $G = (V, E)$, GPEN generates a representation $Z_{S_i}$ for each subgraph to predict its label $y_i$.

**Spatial GCN.** Graph convolution for Spatial GCN is based on the aggregation of node neighbourhood features. Nodes update their features by passing and receiving messages (features). The formulation is:

$$\boldsymbol{h}_v^{(l)} = \mathrm{M}\left(\boldsymbol{h}_v^{(l-1)}, \mathrm{AGG}\{\boldsymbol{h}_u^{(l-1)} : u \in \mathcal{N}(v)\}\right) \quad (1)$$

where $\boldsymbol{h}_v^{(l)}$ represents the feature vector of node $v$ at layer $l$. $\mathcal{N}(v)$ denotes the neighbours of $v$. AGG and M denotes aggregation and update operations, respectively.

**Tree.** Trees are widely used in computer science and mathematics for hierarchical data organization, traversal, and dynamic programming. A tree is a connected acyclic graph commonly represented as $T = (V_T, E_T)$, where $V_T$ is the set of nodes (vertices) and $E_T \subseteq V_T \times V_T$ is the set of edges. In a tree, any two nodes are connected by exactly one simple path, and there are no cycles. A tree with $n$ nodes has exactly $n - 1$ edges.

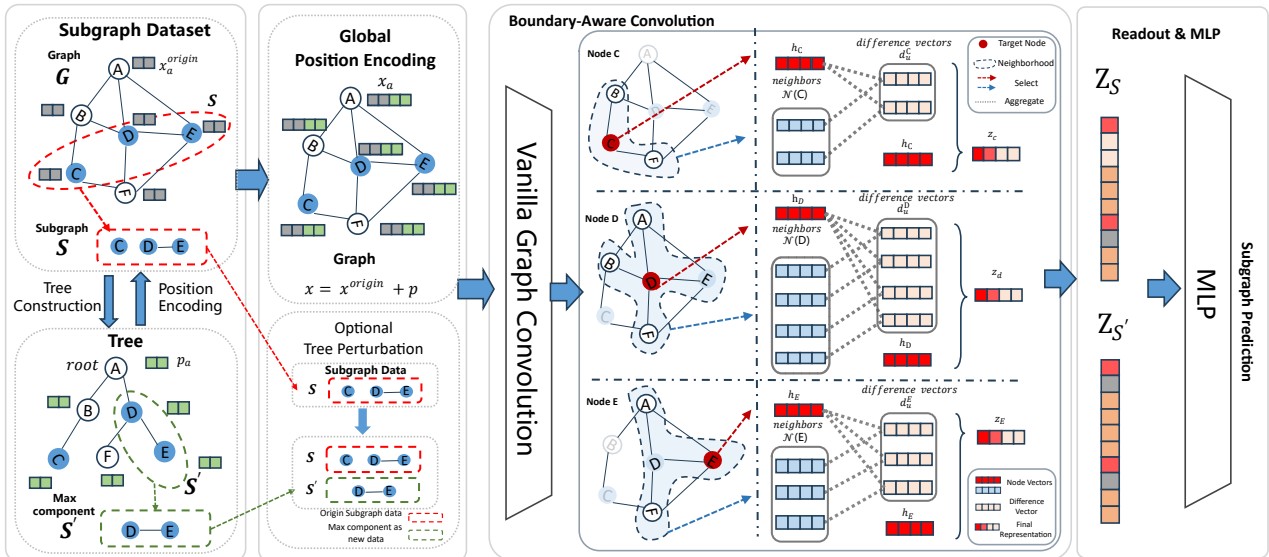

*Figure 2.* Framework of GPEN. (1) **Left Side (Global Position Encoding)**: The input consists of the entire graph $G$ and a subgraph $S = \{C, D, E\}$. GPEN utilizes a tree structure to encode each node's global position, enabling it to capture relationships between distant nodes. Optionally, the largest connected component in the tree can be used for perturbation. (2) **Middle (Boundary-aware Convolution)**: After vanilla message-passing, node representations $h_v$ in the subgraph are refined using difference vectors $d_u^v$ between nodes to selectively integrate global structural information while preserving the subgraph's inherent characteristics. (3) **Right Side (Subgraph Representation & Prediction)**: The node representations are aggregated to obtain the subgraph representation $\{Z_S, Z_{s'}\}$, which is then fed into an MLP to predict the subgraph labels.

## 3. Methodology

In this section, we describe GPEN in detail. GPEN contains two key modules: (1) Global Position Encoding and (2) Boundary-aware Convolution. Global position encoding leverages the natural hierarchical organization of a tree structure to encode each node's global position, enabling a systematic way to capture relationships between distant nodes. Boundary-aware convolution computes difference vectors between nodes to control information flow during convolution, allowing the model to selectively integrate global structural information while maintaining the unique structural patterns of each subgraph.

**Definition 3.1.** Global structural information: For a node $v$ in graph $G$, its global structural information is characterized by its hierarchical position in a tree $T$ constructed from $G$. Formally, let $t_v = d_T(v, r)$ denote the shortest path distance from $v$ to the root node $r$ in $T$. The global structural information of $v$ is encoded through $t_v$, which reflects its structural influence relative to the most central node (root) in the hierarchical organization of $T$.

### 3.1. Global Position Encoding

The global structural information in the entire graph can affect the labels of subgraphs. To acquire global structural information for subgraphs, we introduce the global position of nodes, which takes into account all nodes. To achieve

this, GPEN leverages a tree structure to encode each node's global position, enabling a systematic way to capture relationships between distant nodes. Specifically, GPEN first estimates node importance through degree-based random walks, and then utilizes this information to construct a tree for position encoding. Finally, GPEN encodes each node's global position based on its path distance to the root node.

**Definition 3.2.** Global Position Encoding: Given a tree $T$ with root $r$ and depth $t_{\max}$, the global position encoding of node $v$ is a vector $\boldsymbol{p}_v \in \{0, 1\}^{t_{\max}+1}$ defined as:

$$\boldsymbol{p}_v[i] = \begin{cases} 1 & \text{if } i = t_v \\ 0 & \text{otherwise} \end{cases}, \quad \forall i \in \{0, 1, \ldots, t_{\max}\} \quad (2)$$

where $t_v = d_T(v, r)$ is the shortest path distance from $v$ to $r$ in $T$.

We will explain the process of tree construction and global position encoding.

#### 3.1.1. TREE CONSTRUCTION

Inspired by PageRank (Page et al., 1999), we leverage node degrees to measure node importance, as degrees naturally reflect how well-connected and influential nodes are in the graph (Borgatti, 2005; Lü et al., 2016; Tan et al., 2023). The importance of each node $v_i$ is calculated through iteration based on degrees:

$$R(t+1) = (1-\alpha)MR(t) + \alpha p \qquad (3)$$

where $R(t)$ is the importance vector at iteration $t$, $M$ is the degree-normalized adjacency matrix where $M_{ij} = \frac{1}{d_j}$ if there is an edge between nodes $i$ and $j$ (and $d_j$ is the degree of node $j$), $\alpha$ is the probability of random jumps, and $p$ is the initial probability vector with each element equal to $\frac{1}{|V|}$, where $|V|$ is the number of nodes in the graph.

To utilize the importance values for tree construction, we assign edge weights based on the importance of connected nodes:

$$w_{ij} = R[i] + R[j] \quad (i,j) \in E \qquad (4)$$

The node with the highest importance value is selected as the root node $r$:

$$r = \arg\max_{v \in V} R(v) \qquad (5)$$

Using the edge weights, we construct a weighted graph $G' = (V, E, W)$, where $W$ is the set of weights for the edges. The weighted graph $G'$ and root $r$ allow the use of different algorithms to construct the tree:

$$T = \text{ALGORITHM}(G', r) \qquad (6)$$

where $T$ is the tree, ALGORITHM denotes algorithms for tree construction. For example, using the maximum spanning tree algorithm, the tree is constructed by connecting all nodes using the edges with the highest weights.

### 3.1.2. GLOBAL POSITION ENCODING

The shortest path from every node to the root naturally divides the tree into different levels. This hierarchical structure facilitates the efficient encoding of all nodes. The global positional encoding of a node $v$ can be defined as a one-hot vector representing its shortest path distance $p_v$ from the root node:

$$p_v = [\mathbb{I}(t_v = 0), \mathbb{I}(t_v = 1), ..., \mathbb{I}(t_v = t_{max})] \in \mathbb{R}^{t_{max}} \qquad (7)$$

where $t_v$ is the shortest path distance from the root node, $\mathbb{I}$ is the identity function and $t_{max}$ is the maximum possible value of $t$ equal to the depth of the tree. For the function $\mathbb{I}$, $\mathbb{I}(q) = 1$ when the condition $q = true$, otherwise 0.

Importantly, the global position encoding is computed offline. The final feature $x_i$ for $v_i \in V$ can be obtained by concatenating origin feature $x_i^{origin}$ and $p_v$ together, which will be utilized in subsequent steps of the model.

### 3.1.3. OPTIONAL TREE PERTURBATION

Beyond using the tree structure for global position encoding, we further explore its potential and discover a simple trick

to enhance model robustness. The insufficient number of subgraphs can affect the model's stability. For example, as shown in Table 2, the em-user dataset has 57,333 nodes and 4,573,417 edges, but only 324 subgraphs. This substantial disparity can result in large variations in outcomes under different seeds.

This trick utilizes the tree structure to generate additional samples through consistent perturbation. As shown in the Left Side of Figure 2, for each subgraph $S$, we extract its largest connected component in the tree as a new sample:

$$\begin{cases} S' = f_c(V_S, T) \\ y' = y \end{cases} \quad if \ |V_{S'}| \geq c \qquad (8)$$

where $S'$ and $y'$ are the generated subgraph and its label, $V_S$ is the node set of $S$, and $f_c$ denotes the function that selects the largest connected component of $V_S$ within $T$. The threshold $c$ ensures that only sufficiently large components are used as new samples.

This trick is effective because the tree is constructed by considering the weights of all nodes and edges in the entire graph, preserving much of the original graph's structural information. Moreover, since all perturbations are derived from the same tree structure, they follow a consistent pattern, ensuring the quality and coherence of the generated samples.

Importantly, this simple trick introduces no additional parameters that need careful tuning, as it utilizes the already constructed tree structure. Our experiments show that this optional step can help reduce the variance of results across different random seeds.

### 3.2. Boundary-aware Convolution

Excessive aggregation of global structural information may weaken the discriminative ability of subgraph representations by introducing irrelevant distant nodes.

To address this challenge, we propose a boundary-aware convolution that computes difference vectors between nodes to control information flow during convolution. Specifically, for a node in the subgraph, the module first derives the node feature vectors through vanilla message-passing, then computes the difference vectors between the nodes and their neighbors, and finally aggregates the node feature vectors and the difference vectors to obtain the final node representation. Formally,

$$z_v^{(l)} = \text{AGG}\left(h_v^{(l)}, \{d_u^v : u \in \mathcal{N}(v)\}\right) \qquad (9)$$

where $z_v^{(l)}$ is the final representation of node $v$ at layer $l$, $h_v^{(l)}$ represents the feature vector of $v$ as computed by equation 1, $d_u^v$ denotes the difference vectors in the dimensional space

between $v$ and its neighbour $u$, and AGG is the aggregation operation, such as summation.

The difference vectors $\boldsymbol{d}_u^v$ are formulated as:

$$\boldsymbol{d}_u^v = \boldsymbol{h}_v^{(l)} - \boldsymbol{h}_u^{(l)} \quad u \in \mathcal{N}(v) \tag{10}$$

These difference vectors emphasize the internal structure of subgraphs by capturing the relative differences between neighboring nodes, helping maintain the subgraph's unique structural patterns.

To achieve the convolution, we introduce a diagonal matrix $\boldsymbol{M_s}$ which acts as a binary selector for nodes within subgraphs, with $M_{s_{ii}} = 1$ if node $i$ is part of the subgraph, and 0 otherwise. The representations of graph are denoted as,

$$\boldsymbol{Z}^{(l)} = (1 - b)\boldsymbol{H}^{(l)} + b\boldsymbol{M_s}\boldsymbol{D}^{(l)} \tag{11}$$

where $b$ is a balance factor controlling the integration of global structural information, $\boldsymbol{Z}^{(l)}$ is the representation matrix of nodes, $\boldsymbol{H}^{(l)}$ is the result of vanilla graph convolution and $\boldsymbol{D}^{(l)}$ is the difference matrix. $\boldsymbol{H}$ and $\boldsymbol{D}^{(l)}$ are formulated as:

$$\boldsymbol{H}^{(l)} = \sigma(\boldsymbol{A}\boldsymbol{H}^{(l-1)}\boldsymbol{W}^{(l-1)}) \tag{12}$$

and

$$\boldsymbol{D}^{(l)} = \left[\sum_{u \in \mathcal{N}(v_1)} \mathbf{d}_u^{v_1}, \dots, \sum_{u \in \mathcal{N}(v_n)} \mathbf{d}_u^{v_n}\right]^T \tag{13}$$

where $\boldsymbol{H}^{(0)} = \boldsymbol{X}$. $\boldsymbol{X}$ is the node feature matrix after global position encoding, allowing GPEN to preserve subgraph structures while selectively incorporating global structural information.

## 4. Theoretical Analysis

To theoretically validate GPEN, we provide a comprehensive analysis of its key properties. We first demonstrate that GPEN provides controllable representation learning through bounded representation discrepancy. Then we show how the global position encoding effectively captures structural information. Further analysis proves the stability of boundary-aware convolution against noise propagation, and finally establishes generalization guarantees for the tree perturbation technique.

We begin by showing that GPEN maintains controllable representation learning, with bounded discrepancy between its representations and those of standard GNNs:

**Theorem 4.1** (Bounded Representation Discrepancy). *Let $Z^{GPEN}$ and $Z^{GNN}$ denote subgraph representations generated by GPEN and a standard GNN, respectively. The dis-*crepancy between them is bounded by:

$$\begin{aligned} \|Z^{GPEN} - Z^{GNN}\| \leq &\, C_1 \sum_{v \in V} \|h_v - h_v^{GNN}\| \\ &+ C_2 D(\mathcal{D}^{(l)GPEN}, \mathcal{H}^{GNN}) \\ &+ C_3 D(X^{GPEN}, X^{GNN}). \end{aligned} \tag{14}$$

*where $C_1, C_2, C_3$ are constants dependent on model depth $L$ and Lipschitz continuity of aggregation functions. $X^{GPEN} = [X^{origin}, P]$ and $X^{GNN} = X^{origin}$ are the input features for GPEN and standard GNN respectively, where $P = \{\boldsymbol{p}_v\}_{v \in V}$.*

The proof is provided in the Appendix A.1. This controllability result ensures that GPEN's representations maintain meaningful connections to standard GNN representations while incorporating additional structural information. The bound demonstrates that our extensions provide enhanced expressiveness within controlled limits.

Building on the controllability result, we next analyze whether our global position encoding can effectively distinguish nodes based on their structural roles:

**Theorem 4.2** (Global Position Encoding Distinctness). *Let $G = (V, E)$ be a connected graph, and $T$ be the tree constructed using edge weights $w_{ij} = R(i) + R(j)$, where $R(v)$ is the PageRank score of node $v$. For any two nodes $u, v \in V$ with $\deg(u) \neq \deg(v)$, their global position encodings satisfy $p_u \neq p_v$.*

Full proof is in the Appendix A.2. This theorem confirms that our tree-based global position encoding successfully captures structural differences between nodes, particularly distinguishing nodes with different connectivity patterns.

Having established the effectiveness of position encoding, we analyze the stability of our boundary-aware convolution mechanism:

**Theorem 4.3** (Noise Robustness). *Let $\tilde{h}_v = h_v + \epsilon_v$ be noisy node features with $\epsilon_v \sim \mathcal{N}(0, \sigma^2 I)$. The covariance of boundary-aware convolution outputs satisfies:*

$$Cov(z_v^{BA}) \preceq (1 + 3|\mathcal{N}(v)| + |\mathcal{N}(v)|^2)\sigma^2 I,$$

*while standard GNN aggregation yields $Cov(z_v^{GNN}) = |\mathcal{N}(v)|\sigma^2 I$. Furthermore, boundary-aware convolution achieves higher signal-to-noise ratio (SNR).*

The detailed proof is included in Appendix A.6. This stability analysis reveals that our boundary-aware convolution effectively suppresses noise propagation compared to standard GNN aggregation. The difference vectors used in boundary-aware convolution help maintain structural distinctness while reducing noise amplification, which is essential for robust subgraph representation learning.

| Model | ppi-bp | hpo-metab | hpo-neuro | em-user |
|---|---|---|---|---|
| MLP | $0.445 \pm 0.009$ | $0.386 \pm 0.035$ | $0.404 \pm 0.019$ | $0.524 \pm 0.060$ |
| GBDT | $0.446 \pm 0.000$ | $0.404 \pm 0.000$ | $0.513 \pm 0.000$ | $0.694 \pm 0.000$ |
| Sub2Vec | $0.388 \pm 0.003$ | $0.472 \pm 0.032$ | $0.618 \pm 0.009$ | $0.779 \pm 0.041$ |
| GNN-seg | $0.361 \pm 0.025$ | $0.542 \pm 0.028$ | $0.647 \pm 0.003$ | $0.725 \pm 0.010$ |
| SubGNN | $0.599 \pm 0.025$ | $0.537 \pm 0.025$ | $0.644 \pm 0.019$ | $0.816 \pm 0.041$ |
| GLASS | $0.619 \pm 0.022$ | $0.614 \pm 0.016$ | $0.685 \pm 0.016$ | $0.888 \pm 0.019$ |
| SSNP | $0.636 \pm 0.022$ | $0.587 \pm 0.032$ | $0.682 \pm 0.013$ | $0.888 \pm 0.016$ |
| S2N | $\underline{0.643 \pm 0.041}$ | $\underline{0.639 \pm 0.072}$ | $\underline{0.686 \pm 0.019}$ | $\underline{0.890 \pm 0.035}$ |
| *GPEN* | $\mathbf{0.644 \pm 0.009}$ | $\mathbf{0.639 \pm 0.009}$ | $\mathbf{0.691 \pm 0.006}$ | $\mathbf{0.912 \pm 0.013}$ |

*Table 1.* The mean micro-F1 scores (average of 10 runs) with standard deviations on four real-world datasets.

Finally, we analyze how our optional tree perturbation technique enhances model generalization:

**Theorem 4.4** (Generalization Bound). *Let* $\beta(k) = \sqrt{\mathbb{E}_{S'}[\|S \Delta S'\|]/k}$ *measure perturbation effects. With probability* $1 - \delta$, *the expected risk* $L(h)$ *is bounded by:*

$$L(h) \le L_{emp}(h) + \sqrt{\frac{2 d_{VC} \ln(em/k)}{m}} + \sqrt{\frac{\ln(1/\delta)}{2m}} + \beta(k),$$

*where* $d_{VC}$ *is the VC dimension and* $m, k$ *are the numbers of original and perturbed subgraphs.*

The full proof can be found in Theorem A.7 in the Appendix. This generalization bound demonstrates that tree perturbation improves model generalization by generating structurally consistent augmented samples. The bound quantifies how the additional samples from tree perturbation help reduce overfitting.

The theoretical analysis above establishes four key properties of GPEN: (1) controllable representation learning with bounded discrepancy, (2) effective structural distinction through global position encoding, (3) robust feature learning via boundary-aware convolution, and (4) enhanced generalization through tree perturbation. Together, these properties validate our approach to subgraph representation learning and demonstrate how GPEN addresses the challenges outlined in the introduction.

## 5. Experimental Evaluation

### 5.1. Experimental Settings

#### 5.1.1. DATASETS.

We use the same four real-world datasets and four synthetic datasets (Alsentzer et al., 2020b; Wang & Zhang, 2022; Kim & Oh, 2024). Detailed information about these datasets is presented in Table 2 and Appendix A.2.1. The datasets are divided according to the split ratios outlined in the baselines (Alsentzer et al., 2020b; Wang & Zhang, 2022; Jacob et al., 2023; Kim & Oh, 2024).

| | # nodes | # edges | # Subgraphs | # Classes |
|---|---|---|---|---|
| ppi-bp | 17,080 | 316,951 | 1,591 | 6 |
| hpo-metab | 14,587 | 3,238,174 | 2,400 | 6 |
| hpo-neuro | 14,587 | 3,238,174 | 4,000 | 10 |
| em-user | 57,333 | 4,573,417 | 324 | 2 |
| density | 5,000 | 29,521 | 250 | 3 |
| cut-ratio | 5,000 | 83,969 | 250 | 3 |
| coreness | 5,000 | 118,785 | 221 | 3 |
| component | 19,555 | 43,701 | 250 | 2 |

*Table 2.* Statistics of all datasets.

#### 5.1.2. BASELINES

We compare GPEN with GNN-seg, MLP, GBDT, Sub2Vec (Adhikari et al., 2018), SubGNN (Alsentzer et al., 2020b), GLASS (Wang & Zhang, 2022), SSNP (Jacob et al., 2023), and S2N (Kim & Oh, 2024) as our baselines. For detailed baseline introductions and experimental settings, please refer to Appendix A.2.

### 5.2. Experimental Results

#### 5.2.1. REAL-WORLD DATASETS.

GPEN outperforms all existing models on four real-world datasets. The results are summarized in Table 1, which reports the average micro-F1 scores and standard deviations for each model and dataset (averaged over 10 runs with different seeds). The em-user dataset which contains denser connections and a higher average node degree. GPEN easily outperforms other baselines on the em-user datasets, likely because the larger graph sizes result in deeper trees, making the tree structure more impactful. This result highlights GPEN's superiority in capturing node interaction patterns in a larger graph setting, validating the great performance of GPEN. Notably, simpler baselines like MLP and GBDT that only consider node features perform poorly (0.524 and 0.694 on em-user), while methods that partially capture global structure (GLASS, SSNP) show better results but still fall short of GPEN's comprehensive approach. This empirically validates our theoretical analysis that effective

| Method | density | cut ratio | coreness | component |
|---|---|---|---|---|
| Sub2Vec | $0.459 \pm 0.038$ | $0.354 \pm 0.044$ | $0.360 \pm 0.060$ | $0.657 \pm 0.054$ |
| GNN-seg | $0.952 \pm 0.019$ | $0.346 \pm 0.035$ | $0.593 \pm 0.038$ | $1.000 \pm 0.000$ |
| SubGNN | $0.919 \pm 0.019$ | $0.629 \pm 0.041$ | $0.659 \pm 0.098$ | $0.958 \pm 0.101$ |
| GLASS | $0.930 \pm 0.028$ | $\underline{0.935 \pm 0.019}$ | $\underline{0.840 \pm 0.028}$ | $\underline{1.000 \pm 0.000}$ |
| S2N | $\mathbf{0.963 \pm 0.057}$ | $0.892 \pm 0.044$ | $0.726 \pm 0.114$ | $1.000 \pm 0.000$ |
| *GPEN* | $\underline{0.956 \pm 0.016}$ | $\mathbf{0.936 \pm 0.016}$ | $\mathbf{0.876 \pm 0.019}$ | $\mathbf{1.000 \pm 0.000}$ |

*Table 3.* The mean micro-F1 scores (average of 10 runs) with standard deviations on four synthetic datasets.

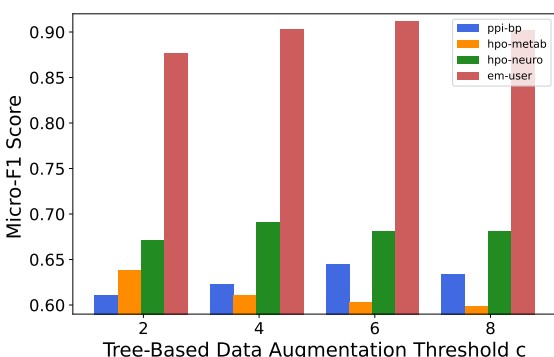

*Figure 3.* Hyperparametric Analysis

| No. | GPE | BWC | OTP | ppi-bp | emuser |
|---|---|---|---|---|---|
| 1 | ✓ | ✓ | ✓ | $0.644 \pm 0.009$ | $0.912 \pm 0.013$ |
| 2 | ✓ | ✓ | | $0.633 \pm 0.019$ | $0.898 \pm 0.032$ |
| 3 | | ✓ | ✓ | $0.590 \pm 0.016$ | $0.796 \pm 0.050$ |
| 4 | ✓ | | ✓ | $0.631 \pm 0.013$ | $0.864 \pm 0.028$ |
| 5 | ✓ | | | $0.632 \pm 0.013$ | $0.857 \pm 0.033$ |
| 6 | | ✓ | | $0.585 \pm 0.028$ | $0.796 \pm 0.054$ |
| 7 | | | ✓ | $0.584 \pm 0.019$ | $0.796 \pm 0.032$ |
| 8 | | | | $0.588 \pm 0.022$ | $0.850 \pm 0.076$ |

*Table 4.* The mean micro-F1 scores and their standard deviations, averaged over 10 runs, for the ablation studies.

subgraph representation requires both global position information and selective feature integration. Finally, GPEN achieves lower standard deviations while maintaining higher accuracy, validating the theoretical stability of our modules.

#### 5.2.2. SYNTHETIC DATASETS.

The synthetic datasets provide controlled experiments to validate specific components of GPEN. As shown in Table 3, GPEN achieves state-of-the-art performance on three out of four datasets, with particularly strong results on tasks requiring global structural understanding.

On the cut-ratio dataset, which tests boundary structure recognition, GPEN achieves 0.936 accuracy with ±0.016 deviation, outperforming other methods and demonstrating the effectiveness of our boundary-aware convolution module. The strong performance on coreness (0.876), which evaluates both boundary and positional information, further validates GPEN's ability to capture multi-hop relationships through global position encoding.

While simpler models like GNN-seg perform well on density and component tasks that primarily test local structure recognition, they struggle with tasks requiring global context (0.346 on cut-ratio). In contrast, GPEN maintains competitive performance on local structure tasks (0.956 on density) while excelling at tasks requiring global information, empirically validating its design goal of balancing local and global structural information.

The consistently lower standard deviations across all

datasets (e.g., ±0.016 on cut-ratio compared to ±0.044 for S2N) demonstrate the stability benefits of our tree-based position encoding and boundary-aware feature integration approach, aligning with our theoretical analysis of the model's representational properties.

#### 5.2.3. ABLATION STUDY

To validate the power of the modules, we conducted ablation studies by removing each corresponding module from GPEN. For convenience, we use abbreviations for the three modules: Global Position Encoding (GPE), Boundary-aware Convolution (BWC), and Optional Tree Perturbation (OTP). The results of the ablation experiments are shown in Table 4. Each module acting individually cannot improve performance and may even worsen it on the emuser dataset. However, when the two modules work together, the results improve, and the best performance is achieved when all three modules are used together. This indicates that the three modules are interdependent. When they work together, they can better utilize both global and local structural information of the subgraphs, achieving optimal performance. It can be observed that removing GPE module leads to a significant performance drop, highlighting the important role of global position encoding in subgraph representation. Additionally, when the OTP module is introduced, GPEN not only shows improved results but also experiences a reduction in standard deviation, demonstrating the effectiveness of the OTP module.

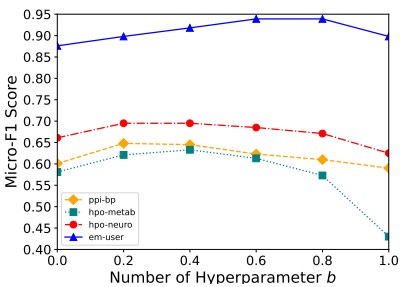

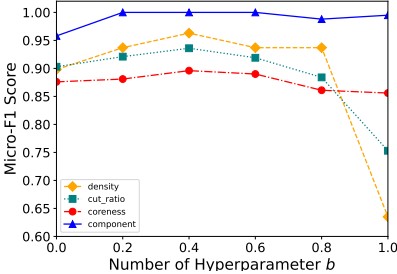

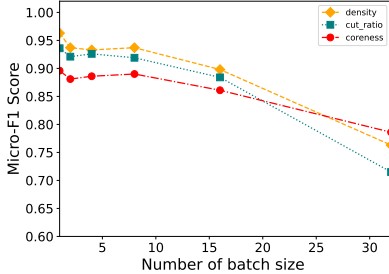

(a) The results of four real-world datasets  (b) The results of four synthetic datasets  (c) The results of different batch sizes

*Figure 4.* Hyperparametric Analysis

| Tree | ppi-bp | hpo-metab | hpo-neuro | em-user |
|---|---|---|---|---|
| Breadth-First Search Tree | 0.560 ± 0.054 | 0.515 ± 0.031 | 0.606 ± 0.037 | 0.857 ± 0.047 |
| Depth-First Search Tree | 0.579 ± 0.047 | 0.494 ± 0.030 | 0.605 ± 0.034 | 0.846 ± 0.035 |
| Minimum Spanning Tree/Steiner Tree | 0.570 ± 0.030 | 0.587 ± 0.026 | 0.683 ± 0.019 | 0.833 ± 0.022 |
| *Maximum Spanning Tree* | **0.644 ± 0.009** | **0.638 ± 0.009** | **0.691 ± 0.006** | **0.912 ± 0.013** |

*Table 5.* Different constructing tree algorithms for the tree

### 5.2.4. TREE ANALYSIS

We evaluated four tree construction algorithms to determine their impact on GPEN's performance: Breadth-First Search Tree (BFS), Depth-First Search Tree (DFS), Minimum Spanning Tree (MST), and Maximum Spanning Tree (MaxST).

Table 5 shows that the maximum spanning tree algorithm achieved the best results. This is likely because we use edge weights to construct the tree, and the maximum spanning tree algorithm focuses on retaining high-weight edges, preserving the more informative structures of the graph.

Specifically, the effectiveness of MaxST stems from its ability to preserve critical structural information by prioritizing connections between high-importance nodes through edge weights $w_{ij} = R[i] + R[j]$, where $R[i]$ represents the PageRank-derived importance score. This approach captures the graph's hierarchical backbone more effectively than traversal-based methods (BFS/DFS) or minimum-weight approaches (MST).

MaxST demonstrates significant performance improvements, particularly in dense graphs such as hpo-metab and hpo-neuro, substantially outperforming BFS and DFS methods. Additionally, MaxST exhibits superior stability with consistently lower standard deviations.

While MaxST has higher computational complexity ($O(|E| \log |V|)$) compared to BFS/DFS ($O(|V| + |E|)$), the significant performance gains justify this cost. The enhanced quality of global position encodings and improved model robustness make MaxST the optimal choice for tree

construction in GPEN.

### 5.2.5. HYPERPARAMETRIC ANALYSIS

We conducted comprehensive experiments to understand the impact of three key hyperparameters in GPEN: the balance factor $b$, the tree-based data augmentation threshold $c$, and the batch size. Our analysis reveals distinct patterns that provide insights into the model's behavior and design principles.

**Tree-based Data Augmentation Threshold** $c$**.** The threshold parameter $c$ determines the minimum size of connected components used for generating augmented samples. Our experiments reveal an interesting trade-off, as shown in Figure 3. Lower thresholds tend to generate numerous but potentially noisy augmented samples, as smaller connected components may not preserve meaningful structural properties of the original subgraphs. Higher thresholds, while ensuring quality, restrict the quantity of augmented data, limiting the regularization benefits. The optimal performance typically occurs at moderate threshold values, where the augmentation process achieves a sweet spot between sample quality and quantity. This pattern is particularly pronounced in datasets with larger average subgraph sizes, where the tree structure provides more opportunities for meaningful augmentation.

**Balance Factor** $b$**.** The balance factor controls the integration of local structural features and global position information during boundary-aware convolution. As illustrated in Figure 4(a), we observe a consistent pattern across all

datasets: optimal performance emerges when $b$ falls within a moderate range, typically between 0.6 and 0.8. This finding validates our theoretical framework that effective subgraph representation requires careful balance between local and global information. When $b$ is too low, the model fails to leverage crucial local structural patterns that define subgraph boundaries. Conversely, excessively high values lead to over-emphasis on local structures, causing the model to lose sight of important global contextual information. The consistency of this optimal range across diverse datasets suggests that this balance is a fundamental requirement rather than a dataset-specific phenomenon.

**Batch Size Effects.** The impact of batch size on model performance, depicted in Figure 4(c), reveals the interplay between optimization dynamics and model convergence. Smaller batch sizes generally lead to better performance, which can be attributed to more frequent parameter updates and increased stochasticity in the optimization process. This stochasticity helps the model escape local minima and explore the parameter space more effectively. As batch size increases, we observe performance degradation, particularly notable in tasks requiring fine-grained structural discrimination. This degradation stems from two factors: reduced gradient update frequency per epoch and the averaging effect of larger batches, which can smooth out important learning signals from individual subgraphs with unique structural patterns.

**Dataset-Specific Sensitivity.** Different datasets exhibit varying degrees of sensitivity to hyperparameter changes, as evidenced in Figure 4(b). Datasets with clear structural boundaries (such as component-based tasks) demonstrate remarkable robustness to hyperparameter variations, maintaining stable performance across wide parameter ranges. In contrast, tasks requiring nuanced integration of multi-scale information (such as coreness prediction) show higher sensitivity. This disparity underscores the importance of our dual-module design: while some tasks primarily benefit from either global position encoding or boundary-aware convolution, complex tasks require precise coordination between both modules, making them more sensitive to the balance factor and other hyperparameters.

## 6. Related Work

### 6.1. Graph Neural Networks.

While traditional deep learning approaches (Schuster & Paliwal, 1997; Krizhevsky et al., 2012) have achieved remarkable success with Euclidean data like images (Cheng et al., 2020) and text (Zheng & Zheng, 2019), numerous real-world applications require modeling data in non-Euclidean graph structures. This necessity led to the development of Graph Neural Networks (GNNs), which have become powerful tools for processing graph-structured data. Various architectures have emerged, including GCN (Kipf & Welling, 2016), GAT (Velickovic et al., 2017), and GraphSage (Hamilton et al., 2017), each offering unique approaches to graph learning.

### 6.2. Subgraph Representation Learning.

The field of subgraph analysis has experienced significant growth (Alsentzer et al., 2020a), with applications ranging from predicting graph evolution (Meng et al., 2018) to enhancing graph classification (Wang et al., 2021). However, the specific challenge of learning subgraph representations remained largely unexplored until recently. A breakthrough came with SubGNN (Alsentzer et al., 2020b), which established the foundations of subgraph representation learning and prediction. The model innovates through its anchor-based information sampling mechanism, which bridges local subgraph structures with global graph context through carefully designed information channels. Building on this foundation, GLASS (Wang & Zhang, 2022) introduced a novel perspective by developing a mask-based approach to distinguish subgraph boundaries. Following these works, researchers have proposed some approaches to balance representational power with computational efficiency. Recent advances include SSNP (Jacob et al., 2023), which improves efficiency through stochastic neighborhood sampling during the readout phase, and S2N (Kim & Oh, 2024), which transforms the subgraph learning problem through an innovative graph coarsening approach that balances computational efficiency with comprehensive structural preservation. The field has also benefited from adapting existing methodologies. For instance, Sub2Vec (Adhikari et al., 2018), originally designed for community detection, demonstrates how random walk sampling and language modeling techniques (Le & Mikolov, 2014) can capture subgraph characteristics.

## 7. Conclusion

In this paper, we present GPEN, a novel method for subgraph representation learning that addresses two key challenges: capturing structural relationships between distant nodes and preventing excessive aggregation of global structural information. Through hierarchical tree-based position encoding, GPEN systematically captures multi-hop relationships between nodes, overcoming the limitations of existing local neighborhood-based approaches. The boundary-aware convolution module selectively integrates global structural information while preserving the intrinsic characteristics of each subgraph, thus effectively balancing the trade-off between global and local structural information. Extensive experiments on eight public datasets validate GPEN's superior performance compared to state-of-the-art methods.

## Acknowledgements

This work was supported by the National Key R&D Program of China (No.31400), NSFC (U2468204), Key R&D and Transformation Plan of Qinghai Province (No.2022-QY-218), Science and Technology Plan of Shenzhen (KCXFZ20211020172544004), the Key R&D project in Hainan province, China (NO.ZDYF2024SHFZ051) and the Major Science and Technology Program of Yazhou Bay Innovation Institute of Hainan Tropical Ocean University(2023CXYZD001).

## Impact Statement

This paper introduces GPEN (Global Position Encoding Network), a framework for enhanced subgraph representation learning. The advancement primarily contributes to the efficiency and effectiveness of graph neural networks, with broad applicability in domains such as fraud detection, biomedical research, and social network analysis. We anticipate that this work will not introduce any negative ethical or social impacts.

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

## A. Appendix

### A.1. Theoretical Analysis

**Theorem A.1** (Controllability of Representation Discrepancy). *Let $S = (V, E, X)$ be a subgraph with $|V| = n$, and let $Z^{GPEN}$ and $Z^{GNN}$ denote the subgraph representations generated by GPEN and a standard GNN, respectively. Assume the following:*

- *$\phi$ is the tree construction function.*

- *$f : \mathbb{R}^d \times \mathbb{R}^d \to \mathbb{R}^d$ is a Lipschitz continuous function with $Lip(f) = \prod_{l=1}^{L} \gamma^{(l)}$, where $\gamma^{(l)}$ is the spectral norm bound of the parameter matrix $W^{(l)}$ at layer $l$.*

- *The message-passing functions $g$ (Boundary-aware Convolution) and $m$ (standard GNN) are Lipschitz continuous with constants $Lip(g)$ and $Lip(m)$, respectively.*

- *All parameter matrices $\{W^{(l)}\}$ satisfy $\|W^{(l)}\|_2 \leq \gamma$ for some $\gamma > 0$.*

*Then, the discrepancy between representations is bounded by:*

$$\|Z^{GPEN} - Z^{GNN}\| \leq C_1 \sum_{v \in V} \|h_v - h_v^{GNN}\| + C_2 D(\mathcal{D}^{(l)GPEN}, \mathcal{H}^{GNN}) + C_3 D(X^{GPEN}, X^{GNN}), \tag{15}$$

*where:*

- *$h_v = \phi(v, G)$ is the global position encoding of node $v$,*

- *$\mathcal{D}^{(l)GPEN} = \{d_u^{(l)} | u \in V\}$ is the multiset of local representations from the $l$-th Boundary-aware Convolution,*

- *$X^{GPEN} = [X^{origin}, P]$ and $X^{GNN} = X^{origin}$ are the input features for GPEN and standard GNN respectively, where $P = \{\boldsymbol{p}_v\}_{v \in V}$,*

- *$D(A, B) = \inf_{\pi \in \Pi} \sup_{a \in A} \|a - \pi(a)\|$ is the optimal matching metric,*

- *Constants $C_1 = L \cdot Lip(f)$, $C_2 = L \cdot d \cdot Lip(f)$, and $C_3 = Lip(f) \cdot Lip(m)$ depend on the model depth $L$ and hidden dimension $d$.*

*Proof.* We decompose the discrepancy via triangle inequality:

$$\|Z^{\text{GPEN}} - Z^{\text{GNN}}\| \leq \underbrace{\left\| f\left(\sum h_v, \sum d_v^{(l)}\right) - f\left(\sum h_v^{\text{GNN}}, \sum d_v^{(l)}\right) \right\|}_{\text{Term 1}}$$
$$+ \underbrace{\left\| f\left(\sum h_v^{\text{GNN}}, \sum d_v^{(l)}\right) - \sum h_v^{\text{GNN}} \right\|}_{\text{Term 2}}. \tag{16}$$

**Bounding Term 1:** By Lipschitz continuity of $f$ and the norm inequality:

$$\text{Term 1} \leq \text{Lip}(f) \left( \left\| \sum (h_v - h_v^{\text{GNN}}) \right\| + \left\| \sum d_v^{(l)} \right\| \right)$$
$$\leq \text{Lip}(f) \left( \sum \|h_v - h_v^{\text{GNN}}\| + \sum \|d_v^{(l)}\| \right) \quad \text{(by triangle inequality)}$$
$$\leq \text{Lip}(f) \left( \sum \|h_v - h_v^{\text{GNN}}\| + n \cdot \max_v \|d_v^{(l)}\| \right). \tag{17}$$

Using the optimal matching metric $D(\cdot, \cdot)$, we have:

$$\max_v \|d_v^{(l)}\| \leq D(\mathcal{D}^{(l)\text{GPEN}}, \mathcal{H}^{\text{GNN}}). \tag{18}$$

**Bounding Term 2:** By Lipschitz continuity of $f$ and $m$:

$$\text{Term 2} \leq \text{Lip}(f) \cdot \left\| \sum d_v^{(l)} - \sum m(\sum \alpha_{vw} x_w) \right\|$$
$$\leq \text{Lip}(f) \cdot \text{Lip}(m) \cdot D(X^{\text{GPEN}}, X^{\text{GNN}}). \tag{19}$$

**Combining bounds:**

$$\|Z^{\text{GPEN}} - Z^{\text{GNN}}\| \leq \text{Lip}(f) \sum \|h_v - h_v^{\text{GNN}}\| + \text{Lip}(f) \cdot n \cdot D(\mathcal{D}^{(l)\text{GPEN}}, \mathcal{H}^{\text{GNN}})$$
$$+ \text{Lip}(f) \cdot \text{Lip}(m) \cdot D(X, X)$$
$$= C_1 \sum \|h_v - h_v^{\text{GNN}}\| + C_2 D(\mathcal{D}^{(l)\text{GPEN}}, \mathcal{H}^{\text{GNN}}) + C_3 D(X^{\text{GPEN}}, X^{\text{GNN}}), \tag{20}$$

where $C_1 = L \cdot \text{Lip}(f)$, $C_2 = L \cdot d \cdot \text{Lip}(f)$, and $C_3 = \text{Lip}(f) \cdot \text{Lip}(m)$, incorporating the model depth $L$ and hidden dimension $d$. $\qquad\square$

**Theorem A.2** (Global Position Encoding Distinctness with Tree Construction Constraints). *Let $G = (V, E)$ be a connected graph and $u, v \in V$ with $\deg(u) \neq \deg(v)$. Let $T$ be constructed via using maximum spanning tree (MST) with edge weights $w_{ij} = R(i) + R(j)$, where $R(v)$ is the PageRank score of node $v$. Assuming node importance $R(v)$ is strictly increasing with degree (i.e., $\deg(u) > \deg(v) \implies R(u) > R(v)$), then their global position encodings satisfy $p_u \neq p_v$.*

*Proof.* We first establish two lemmas:

**Lemma A.3** (Monotonicity of PageRank). *Under the iteration in Eq.(3) with $\alpha \in (0, 1)$, for connected graphs, $\deg(u) > \deg(v) \implies R(u) > R(v)$.*

*Proof.* From the PageRank equation $R = (1 - \alpha)MR + \alpha p$, where $M_{ij} = 1/\deg(i)$. For node $u$ with higher degree:

$$R(u) = (1 - \alpha) \sum_{j \in N(u)} \frac{R(j)}{\deg(j)} + \frac{\alpha}{|V|}$$

By the Perron-Frobenius theorem, stationary distribution components satisfy $R(u)/\deg(u) > R(v)/\deg(v)$ when $\deg(u) > \deg(v)$. Hence $R(u) > R(v)$. $\qquad\square$

**Lemma A.4** (MST Path Property). *In MST $T$ with edge weights $w_{ij} = R(i) + R(j)$, the path from any node $v$ to root $r$ is monotonically increasing in $R(\cdot)$ values.*

*Proof.* Suppose there exists a decreasing edge $(x, y)$ on the path with $R(x) > R(y)$. Then replacing it with edge $(y, z)$ where $R(z) > R(y)$ would increase the total weight, contradicting MST maximality. $\qquad\square$

Now consider $u, v$ with $\deg(u) > \deg(v)$. By Lemma A.3, $R(u) > R(v)$. Let $r =_v R(v)$. There are two cases:

**Case 1:** $v$ lies on the path from $u$ to $r$ in $T$. By Lemma A.4, $\text{dist}_T(u, r) = \text{dist}_T(v, r) + \text{dist}_T(u, v) > \text{dist}_T(v, r)$.

**Case 2:** $u$ and $v$ belong to different subtrees. Let $w$ be the lowest common ancestor. Then:

$$\text{dist}_T(u, r) = \text{dist}_T(u, w) + \text{dist}_T(w, r) \neq \text{dist}_T(v, w) + \text{dist}_T(w, r) = \text{dist}_T(v, r)$$

since $u$ and $v$ cannot have identical distances to $w$ while maintaining MST maximality under weight ordering.

Thus $p_u \neq p_v$ in all cases. $\qquad\square$

**Corollary A.5** (Degree-Induced Structural Distinctness). *Let $\mathcal{I}(v) = t_v$ denote the global structural information of node $v$ defined as its depth in $T$. For any $u, v \in V$ with $\deg(u) \neq \deg(v)$, their structural information satisfies $\mathcal{I}(u) \neq \mathcal{I}(v)$. The converse does not hold: $\mathcal{I}(u) \neq \mathcal{I}(v)$ may occur even when $\deg(u) = \deg(v)$.*

*Proof.* The first statement follows directly from Theorem A.2. For the converse, consider a graph with two 3-degree nodes $u$ and $v$ where $u$ connects to leaf nodes while $v$ connects to hub nodes. Their MST depths may differ despite equal degrees. $\qquad\square$

**Theorem A.6** (Stability of Boundary-aware Convolution). *Let $\tilde{h}_v = h_v + \epsilon_v$ be the observed node features with additive noise $\epsilon_v \sim \mathcal{N}(0, \sigma^2 I)$, where $\{\epsilon_v\}_{v \in V}$ are mutually independent. Given a subgraph $S \subseteq G$, define the following two aggregation operators:*

- *Standard GNN aggregation: $z_v^{GNN} = \sum_{u \in \mathcal{N}(v)} \tilde{h}_u$*

- *Boundary-aware aggregation: $z_v^{BA} = \tilde{h}_v + \sum_{u \in \mathcal{N}(v)} (\tilde{h}_v - \tilde{h}_u)$*

*Then the covariance matrices satisfy:*

$$\mathrm{Cov}(z_v^{BA}) = (1 + 3|\mathcal{N}(v)| + |\mathcal{N}(v)|^2)\sigma^2 I \quad and \quad \mathrm{Cov}(z_v^{GNN}) = |\mathcal{N}(v)|\sigma^2 I$$

*Furthermore, the signal-to-noise ratio (SNR) satisfies:*

$$\frac{\mathbb{E}[\|z_v^{BA}\|_2^2]}{\mathrm{Tr}(\mathrm{Cov}(z_v^{BA}))} \geq \frac{\mathbb{E}[\|z_v^{GNN}\|_2^2]}{|\mathcal{N}(v)|\sigma^2 d}$$

*where $d$ is the feature dimension.*

*Proof.* We first analyze the boundary-aware aggregation. Let $d_u^v = \tilde{h}_v - \tilde{h}_u = (h_v - h_u) + (\epsilon_v - \epsilon_u)$. The aggregation becomes:

$$z_v^{BA} = h_v + \underbrace{\sum_{u \in \mathcal{N}(v)} (h_v - h_u)}_{\text{signal term}} + \underbrace{\epsilon_v + \sum_{u \in \mathcal{N}(v)} (\epsilon_v - \epsilon_u)}_{\text{noise term}}$$

The noise covariance is:

$$\mathrm{Cov}(z_v^{BA}) = \mathrm{Cov}\left( \epsilon_v + \sum_u (\epsilon_v - \epsilon_u) \right) = \mathrm{Cov}\left( (1 + |\mathcal{N}(v)|)\epsilon_v - \sum_u \epsilon_u \right)$$

Using independence of $\{\epsilon_v\}$:

$$= (1 + |\mathcal{N}(v)|)^2 \sigma^2 I + |\mathcal{N}(v)|\sigma^2 I = [1 + 2|\mathcal{N}(v)| + |\mathcal{N}(v)|^2 + |\mathcal{N}(v)|]\sigma^2 I$$

$$= (1 + 3|\mathcal{N}(v)| + |\mathcal{N}(v)|^2)\sigma^2 I$$

For standard GNN aggregation:

$$\mathrm{Cov}(z_v^{GNN}) = \sum_{u \in \mathcal{N}(v)} \mathrm{Cov}(\epsilon_u) = |\mathcal{N}(v)|\sigma^2 I$$

The SNR comparison follows from:

$$\frac{\mathbb{E}[\|z_v^{BA}\|_2^2]}{\mathrm{Tr}(\mathrm{Cov}(z_v^{BA}))} \geq \frac{\|\mathbb{E}[z_v^{BA}]\|_2^2}{(1 + 3|\mathcal{N}(v)| + |\mathcal{N}(v)|^2)\sigma^2 d} \geq \frac{\|\mathbb{E}[z_v^{GNN}]\|_2^2}{|\mathcal{N}(v)|\sigma^2 d}$$

where the first inequality uses Jensen's inequality and the second follows from $\|\mathbb{E}[z_v^{BA}]\|_2^2 = \|h_v + \sum(h_v - h_u)\|_2^2 \geq \|\sum h_u\|_2^2 = \|\mathbb{E}[z_v^{GNN}]\|_2^2$ by triangle inequality and the fact that $(1 + 3|\mathcal{N}(v)| + |\mathcal{N}(v)|^2) \geq |\mathcal{N}(v)|$ for $|\mathcal{N}(v)| \geq 1$. $\square$

**Theorem A.7** (Generalization Bound with Tree Perturbation). *Let $\mathcal{H}$ be the hypothesis class with VC dimension $d_{VC}$ (Vapnik & Chervonenkis, 2015), trained on $m$ original subgraphs $S_1, \ldots, S_m$ and $k$ perturbed subgraphs $S_1', \ldots, S_k'$ generated via tree perturbation. Let $\mathcal{L}_{emp}(h)$ and $\mathcal{L}(h)$ denote the empirical and expected risks respectively. With probability at least $1 - \delta$ over the sample generation, for any $h \in \mathcal{H}$:*

$$\mathcal{L}(h) \leq \mathcal{L}_{emp}(h) + \sqrt{\frac{2d_{VC}\ln(em/k)}{m}} + \sqrt{\frac{\ln(1/\delta)}{2m}} + \beta(k)$$

*where $\beta(k) = \sqrt{\frac{\mathbb{E}_{S'}[\|S \triangle S'\|]}{k}}$ measures the perturbation effect, and $\|S \triangle S'\|$ is the symmetric difference between original and perturbed subgraphs.*

*Proof.* We proceed through three key steps:

**Lemma A.8** (Rademacher Complexity with Augmentation). *Let $\mathfrak{R}_m(\mathcal{H})$ be the Rademacher complexity ([Bousquet et al., 2003](#)) of $\mathcal{H}$ on original samples. The augmented complexity satisfies:*

$$\mathfrak{R}_{m+k}(\mathcal{H}) \leq \frac{m}{m+k}\mathfrak{R}_m(\mathcal{H}) + \frac{k}{m+k}\left(\mathfrak{R}_k(\mathcal{H}) + \sqrt{\frac{\mathbb{E}[\|S \triangle S'\|]}{2k}}\right)$$

*Proof.* Using the contraction property of Rademacher complexity:

$$\mathfrak{R}_{m+k} \leq \mathbb{E}\left[\sup_h \left(\frac{1}{m}\sum_{i=1}^{m}\sigma_i h(S_i) + \frac{1}{k}\sum_{j=1}^{k}\sigma'_j h(S'_j)\right)\right]$$

Decompose into original and perturbed terms, then apply Talagrand's concentration inequality on the perturbation term ([Boucheron et al., 2003](#)). □

**Lemma A.9** (VC Dimension Preservation). *The tree perturbation process preserves the VC dimension:*

$$d_{VC}(\mathcal{H}') \leq 2d_{VC}(\mathcal{H})$$

*where $\mathcal{H}'$ is the augmented hypothesis space.*

*Proof.* Each perturbation can be viewed as a restriction operator $f_c$ acting on the original space. By the Sauer-Shelah lemma, the growth function of the composed space satisfies $\Pi_{\mathcal{H}'}(n) \leq \Pi_{\mathcal{H}}^2(n)$. □

**Lemma A.10** (Stability of Perturbation). *The expected symmetric difference satisfies:*

$$\mathbb{E}[\|S \triangle S'\|] \leq \mathbb{E}[|V_S| - c]\mathbb{I}_{\{|V_S|>c\}}$$

*where $c$ is the size threshold in Eq.(8).*

*Proof.* By the perturbation definition $S' = f_c(V_S, T)$, the difference comes from nodes outside the largest connected component. Apply linearity of expectation over the node selection process. □

Combining these results via PAC-Bayes framework ([Catoni, 2007](#)):

$$\mathcal{L}(h) \leq \mathcal{L}_{\text{emp}}(h) + 2\mathfrak{R}_{m+k}(\mathcal{H}) + \sqrt{\frac{\ln(1/\delta)}{2(m+k)}}$$

Substitute Lemma A.8 and bound $\mathfrak{R}_k(\mathcal{H})$ using Lemma A.9 through Dudley entropy integral:

$$\mathfrak{R}_k(\mathcal{H}) \leq C\sqrt{\frac{d_{\text{VC}}\ln(ek/d_{\text{VC}})}{k}}$$

Finally apply Lemma A.10 to bound the perturbation term $\beta(k)$. □

## A.2. Experimental Details

### A.2.1. DATASETS

(1) **real-world datasets.** The **ppi-bp** dataset is a molecular biology dataset for predicting cellular functions of protein groups involved in common biological processes. The **hpo-metab** and **hpo-neuro** datasets are clinical diagnostic datasets for predicting the type of rare metabolic or neurological disorder based on phenotype and genotype data from a knowledge graph. The **em-user** dataset is a user profiling dataset for predicting user characteristics, such as gender, based on their workout history within a social fitness network.

(2) **synthetic datasets.** These four synthetic datasets are constructed by generating specific structured subgraphs on base graphs. The labels for each dataset are determined by binning the subgraph specific structure ([Alsentzer et al., 2020b](#)). Detailed information about these datasets is presented in Table 6.

| Dataset | Nodes | Edges | Subgraphs | Average Nodes per Subgraph |
|---------|-------|-------|-----------|----------------------------|
| Density | 5,000 | 29,521 | 250 | $20.0 \pm 0.0$ |
| Cut Ratio | 5,000 | 83,969 | 250 | $20.0 \pm 0.0$ |
| Coreness | 5,000 | 118,785 | 221 | $20.0 \pm 0.0$ |
| Component | 19,555 | 43,701 | 250 | $74.2 \pm 52.8$ |
| PPI-BP | 17,080 | 316,951 | 1,591 | $10.2 \pm 10.5$ |
| HPO-METAB | 14,587 | 3,238,174 | 1,400 | $14.4 \pm 6.2$ |
| HPO-NEURO | 14,587 | 3,238,174 | 4,000 | $14.8 \pm 6.5$ |
| EM-USER | 57,333 | 4,573,417 | 324 | $155.4 \pm 100.2$ |

*Table 6.* Subgraph Properties in Synthetic and Real-World Datasets

### A.2.2. BASELINES

To prove our model validity, we compare GPEN with the following baselines:

(1) **SubGNN** (Alsentzer et al., 2020b) utilizes subgraph-level message passing with artificial channels;

(2) **GLASS** (Wang & Zhang, 2022) employs a label trick with a mask matrix to differentiate between subgraph internal and external regions;

(3) **SSNP** (Jacob et al., 2023) introduces a Stochastic Subgraph Neighborhood Pooling strategy, sampling neighboring nodes during subgraph readout;

(4) **S2N** (Kim & Oh, 2024) translates subgraphs to nodes using graph coarsening methods, while capturing both local and global structures of the subgraph.

(5) **Sub2Vec** (Adhikari et al., 2018) generates subgraph embeddings by sampling random walks and using Paragraph2Vec;

(6) **GNN-seg** is a standard MPNN that performs graph classification by treating each subgraph as an independent graph, ignoring its external connections; and (7) **MLP** and (8) **GBDT** aggregate node embeddings for subgraph classification, disregarding the overall graph structure.

### A.2.3. IMPLEMENTATION DETAILS

we set the number of iterations $t$ for PageRank to 100, with the damping factor $a$ set to 0.85. Similar to GLASS and SubGNN, our model pre-trains nodes to generate node features for real-world datasets. Additionally, global position encoding is added as the initial feature for all datasets. We calculate the vector differences of node representations using a COO-formatted sparse adjacency matrix A, which significantly reduces memory usage. We use classic loss functions for classification tasks: BCE loss for binary classification and cross-entropy loss for multi-class classification.

---

**Algorithm 1** Global Position Encoding

---

**Input**: Graph $G = (V, E)$
**Output**: Global Position Encodings $\{\boldsymbol{p}_v\}_{v \in V}$

1: $\boldsymbol{R} \leftarrow Eq.(3)$
2: $w_{ij} \leftarrow \boldsymbol{R}[i] + \boldsymbol{R}[j], \forall (i,j) \in E$
3: $r \leftarrow \arg\max_{v \in V} \boldsymbol{R}[v]$
4: $G' \leftarrow (V, E, W), W = \{w_{ij}\}$
5: $T \leftarrow Eq.(6)$
6: **for all** $v \in V$ **do**
7: $\quad t_v \leftarrow \text{dist}_T(v, r)$
8: $\quad \boldsymbol{p}_v \leftarrow \text{one\_hot}(t_v)$
9: **end for**$\{\boldsymbol{p}_v\}_{v \in V}$

---

