# OpenReview forum: "GPEN: Global Position Encoding Network for Enhanced Subgraph Representation Learning"
_ICML.cc/2025/Conference — ICML 2025 poster_

### Official Review · Reviewer_QQCz · 2025-02-19

**Overall Recommendation:** 4

**Summary:**

The paper introduces GPEN (Global Position Encoding Network), a novel algorithm for subgraph representation learning. The algorithm addresses the task of predicting labels for subgraphs within a large input graph, given a set of labeled subgraphs. GPEN innovates by moving beyond the limitations of existing methods that primarily focus on local neighborhood structures. It achieves this by incorporating global structural information through a novel tree-based global position encoding for each node. This encoding is combined with a boundary-aware convolution module and an optional tree perturbation technique. The authors support their approach with both empirical evaluations demonstrating effectiveness and theoretical analyses to justify the method.

**Claims And Evidence:**

The paper's main claims are well supported by convincing evidence. Experimental results show that GPEN outperforms existing methods across various datasets, while theoretical analysis validates the method's soundness from multiple angles. In particular, Figure 1 clearly demonstrates the limitations of relying solely on local structural information and how GPEN addresses this issue by capturing global structural information.

**Essential References Not Discussed:**

I did not find any essential references that were missing or overlooked in the paper's discussion of related work.

**Experimental Designs Or Analyses:**

The experimental design is sound and effective. Extensive experiments on eight public datasets, with results averaged over 10 different random seeds, provide reliable statistical significance. The hyperparameter analysis and ablation studies effectively validate the contribution of each module.

**Methods And Evaluation Criteria:**

The proposed methods are innovative and well-justified. GPEN cleverly utilizes tree structures to encode global position information, while the boundary-aware convolution module effectively balances local and global information. The experimental evaluation employs widely recognized datasets and baseline methods in the field, with a well-designed and comprehensive evaluation protocol.

**Other Comments Or Suggestions:**

1. Although the paper includes hyperparameter analysis, a sentence or two summarizing the general robustness or sensitivity of the key hyperparameters within the main body (perhaps in the experimental section) would be helpful for readers. This would provide a concise summary without requiring a deep dive into the appendix.

2. Adding visualizations of the final learned embeddings could highlight the effectiveness of the approach to a reader.

**Other Strengths And Weaknesses:**

Strengths：

The core idea of tree-based global position encoding is novel and effective.
Robust theoretical analysis strengthens the paper.
The paper is well-structured and easy to follow.

Weaknesses:

Visualizations of learned embeddings could further illustrate the effectiveness of GPEN.

**Questions For Authors:**

The current work primarily addresses subgraph representation learning on undirected graphs. Could the authors discuss the applicability of GPEN to directed graphs (e.g., follow relationships in social networks, citation links in academic networks)? Specifically, we are interested in how edge directionality would be handled in the tree construction phase and what modifications to the maximum spanning tree algorithm would be necessary to accommodate directed scenarios. The answer would help understand GPEN's potential for broader applications in directed graph domains.

**Relation To Broader Scientific Literature:**

The work is well-connected with existing research. The paper clearly identifies limitations in existing methods and proposes effective solutions. GPEN's main innovations - global position encoding and boundary-aware convolution - represent significant improvements and additions to prior work.

**Theoretical Claims:**

I carefully checked the correctness of the theoretical claims, including the theorems and proofs related to bounded representation discrepancy, global position encoding distinctness, noise robustness, and the generalization bound. They are robust and clearly presented, providing a strong theoretical foundation for the GPEN algorithm. The inclusion of these formal analyses significantly strengthens the paper.

---

> ### Author Rebuttal · Authors · 2025-03-31
>
> We deeply appreciate your thoughtful review and positive assessment of our work. Below we will respond to the raised questions.
>
> ## W1 and S2: Visualization of Learned Embeddings
>
> We appreciate your suggestion about visualization. We agree that adding visualizations of the learned embeddings would enhance the interpretability of our approach. In the revised manuscript, we will add t-SNE visualizations of subgraph representations learned by GPEN compared with baseline methods on selected datasets. These visualizations will help readers intuitively understand how our method better separates subgraphs of different classes in the embedding space.
>
> ## S1: Hyperparameter Analysis Summary
>
> Thank you for this valuable suggestion. We will add a concise summary of our hyperparameter analysis in Section 5.2.5 of the main experimental section. The key findings include:
>
> - The balance factor b (controlling the trade-off between local and global structural information) shows optimal performance in the range of 0.6-0.8 across all datasets, indicating that moderately emphasizing local structural information while maintaining global context leads to better subgraph representations.
>
> - The tree-based data augmentation threshold c achieves optimal results with moderate values (4-6), suggesting that this range provides an effective balance between generating sufficient augmented samples and maintaining structural integrity.
>
> - Batch size exhibits relatively stable performance for values between 5 and 15, with gradual decline for larger values (>20), likely due to reduced gradient update frequency and less effective early stopping with larger batches.
>
> - GPEN demonstrates reasonable robustness to hyperparameter changes within these ranges, with different datasets exhibiting varying sensitivities based on their inherent structural properties.
>
> ## Q1: Applicability to Directed Graphs
>
> GPEN can be naturally extended to directed graphs. Below we elaborate on how each module can be adapted:
>
> Our importance calculation already accommodates directed graphs, as the transition matrix $M$ inherently represents directional flow where $M_{ij}$ is defined based on the out-degree of nodes. This formulation naturally captures the directional influence of nodes in the graph without requiring fundamental changes.
>
> For tree construction, we would maintain the same edge weight assignment while selecting an appropriate spanning arborescence algorithm (directed version of spanning tree) such as Edmonds' algorithm to construct the tree structure from the weighted directed graph. The root node selection would remain based on the highest PageRank score, preserving our hierarchical encoding approach.
> Once the directed tree (arborescence) is constructed, the global position encoding process remains identical to the undirected case. We would still compute each node's position based on its path distance to the root node, enabling the same systematic way to capture relationships between distant nodes in directed scenarios.
> For boundary-aware convolution, the primary change would be in the adjacency matrix representation, which would need to reflect the directionality of edges. This modification preserves our boundary-aware convolution's ability to control information flow during the process.
>
> These adaptations would enable GPEN to effectively capture hierarchical relationships in directed network scenarios such as citation networks, social influence networks, and information flow systems, while preserving the theoretical guarantees established in our paper.

---

### Official Review · Reviewer_jSEe · 2025-03-09

**Overall Recommendation:** 3

**Summary:**

This paper presents GPEN, a novel method for subgraph representation learning that addresses two key challenges: capturing structural relationships between distant nodes and preventing excessive aggregation of global structural information.

**Claims And Evidence:**

yes, the submission is supported by clear and convincing evidence

**Essential References Not Discussed:**

No

**Experimental Designs Or Analyses:**

yes, the experiment section is well-designed.

**Methods And Evaluation Criteria:**

yes, the proposed method makes sense for the graph learning challenges.

**Other Comments Or Suggestions:**

Please answer W1-W3

**Other Strengths And Weaknesses:**

**Strengths:**

S1. The paper provides rigorous theoretical analysis (Theorems 4.1–4.4) to validate its claims, such as bounded representation discrepancy and noise robustness, which adds intellectual depth to the innovation.

S2. The introduction provides a clear overview of the limitations inherent in existing methods, which highlights the key challenges capturing distant relationships, avoiding over-aggregation. Figure 1 enhances this by visually contrasting fraudulent and legitimate subgraphs, making the motivation intuitive.


**Weaknesses:**


W1. Although the tree-based encoding approach introduces an innovative perspective, its fundamental assumption may limit its applicability, such as a tree structure can sufficiently capture the intricacies of complex graph topologies. Specifically, in graphs characterized by high cyclicity or dense interconnectivity, the hierarchical simplification imposed by tree-based representations could lead to an oversimplification of relational patterns, potentially compromising the model's ability to generalize effectively.

W2. While it is appropriate to provide the detailed theoretical proofs in appendix, it would be beneficial to provide a concise summary of the key insights in the main text, such as the mechanism by which boundary-aware convolution enhances the signal-to-noise ratio. This approach would significantly improve the accessibility and clarity of the work for readers.

W3. There is little discussion of scenarios where it might underperform (e.g., sparse graphs, small subgraphs). This could highlight limitations and guide future work.

Typo: Line 804 'real-work datasets' -> 'real-world datasets'

**Questions For Authors:**

Please answer W1-W3

**Relation To Broader Scientific Literature:**

This paper addresses existing challenges in a more novel way

**Theoretical Claims:**

yes, the theoretical proof of the paper seems correct.

---

> ### Author Rebuttal · Authors · 2025-03-31
>
> We deeply appreciate your kind words regarding the clarity of our presentation. Thank you for acknowledging the thoroughness of our theoretical and experimental sections. Below, we have responded to the weaknesses raised by the reviewer:
>
> **W1: Concerns about tree-based encoding and complex graph topologies**
>
> We appreciate this thoughtful observation. Our extensive experiments were conducted on eight public benchmark datasets that have been widely used in previous subgraph representation learning studies [1-4]. We evaluated GPEN on eight public datasets comprising various graph structures, including dense networks and small subgraphs. For clearer perspective, here's a summary of these datasets:
>
> | Dataset | Nodes | Edges | Avg. Degree | Subgraphs | Avg. Nodes per Subgraph |
> |---------|-------|-------|-------------|-----------|-------------------------|
> | Density | 5,000 | 29,521 | 5.90 | 250 | 20.0 ± 0.0 |
> | Cut Ratio | 5,000 | 83,969 | 16.79 | 250 | 20.0 ± 0.0 |
> | Coreness | 5,000 | 118,785 | 23.76 | 221 | 20.0 ± 0.0 |
> | Component | 19,555 | 43,701 | 2.23 | 250 | 74.2 ± 52.8 |
> | PPI-BP | 17,080 | 316,951 | 18.56 | 1,591 | 10.2 ± 10.5 |
> | HPO-METAB | 14,587 | 3,238,174 | 222.0 | 1,400 | 14.4 ± 6.2 |
> | HPO-NEURO | 14,587 | 3,238,174 | 222.0 | 4,000 | 14.8 ± 6.5 |
> | EM-USER | 57,333 | 4,573,417 | 79.77 | 324 | 155.4 ± 100.2 |
>
> Notably, GPEN achieves superior performance (0.912 micro-F1) on EM-USER, which contains over 4.5 million edges, demonstrating that our approach effectively captures complex structural information in densely connected graphs.
>
> This effectiveness primarily stems from the Maximum Spanning Tree (MaxST) methodology, which preserves critical structural information during tree construction by leveraging node importance scores derived from PageRank. The weighting scheme ensures connections between high-importance nodes are prioritized in the resulting tree structure, effectively capturing the backbone of the graph's hierarchical organization. Our tree analysis (Table 6) shows MaxST consistently outperforms other tree construction methods across all datasets, confirming its ability to capture essential graph topology even when simplifying complex structures. As other reviewers noted, our approach "effectively clarifies the practical significance of utilizing a hierarchical tree in the study."
>
> [1] Alsentzer et al., "Subgraph Neural Networks", NeurIPS 2020
>
> [2] Wang & Zhang, "Glass: GNN with labeling tricks for subgraph representation learning", ICLR 2022
>
> [3] Jacob et al., "Stochastic subgraph neighborhood pooling for subgraph classification", CIKM 2023
>
> [4] Kim & Oh, "Translating subgraphs to nodes makes simple GNNs strong and efficient for subgraph representation learning", ICML 2024
>
> **W2: Request for concise theoretical summaries in main text**
>
>
> We appreciate this constructive suggestion. Our theoretical analysis establishes four key properties:
> 1. Theorem 4.1 proves the controllability of GPEN's representations
> 2. Theorem 4.2 proves the distinctiveness of global position encoding
> 3. Theorem 4.3 proves that the boundary aware convolution module can suppress noise propagation
> 4. Theorem 4.4 proves that the empirical distribution covers a wider range after perturbation.
>
> In our revised manuscript, we will add concise summaries after each theorem in Section 4, highlighting key insights in accessible language and demonstrating how these theoretical properties address the challenges outlined in our introduction.
>
> **W3: Discussion of potential underperformance scenarios**
>
> We thank the reviewer for this valuable suggestion. Regarding synthetic datasets with small subgraphs (Table 3), these were specifically designed to test different capabilities. For density and component tasks, where GNNs already perform well, our model complements these strengths. As shown in Figure 2, our position encoding supplements original features rather than replacing them, while our boundary-aware convolution prevents excessive aggregation. For cut-ratio and coreness datasets, which require more sophisticated structural understanding, our approach effectively captures the necessary information.
>
> Our experiments do reveal some insights about potential limitations. In the Component dataset (Table 3), which has the lowest average degree (2.23) among all datasets, multiple methods including GPEN achieve perfect scores, suggesting that for very sparse graphs with clear component structures, simpler methods may be equally effective. Additionally, our hyperparameter analysis (Figure 4) shows varying sensitivities across datasets, with Coreness exhibiting more pronounced performance fluctuations with changes in the balance factor b, indicating that optimal parameter tuning may be more critical for certain graph structures.
>
>
> **Typo correction**
>
> Thank you for pointing out the typo on line 804. We will correct "real-work datasets" to "real-world datasets" in the revised manuscript.

---

### Official Review · Reviewer_2fVD · 2025-03-12

**Overall Recommendation:** 4

**Summary:**

This paper presents GPEN (Global Position Encoding Network), a novel approach for subgraph representation learning that addresses the limitation of existing methods which primarily focus on local neighborhood structures while overlooking global structural information. GPEN implements two key modules: (1) global position encoding, which leverages hierarchical tree structure to encode each node's global position, enabling the capture of structural relationships between distant nodes; and (2) boundary-aware convolution, which computes difference vectors between nodes to control information flow, selectively integrating global structural information while maintaining the unique structural patterns of each subgraph. Experiments show that GPEN achieves competitive or superior performance compared to state-of-the-art methods on eight public datasets.

**Claims And Evidence:**

The main claims of the paper are well-supported by theoretical analysis and experimental results. The authors claim that GPEN can effectively capture global structural information while preserving the unique features of subgraphs, which is supported by multiple theoretical analyses and relatively comprehensive experimental validation. Theoretically, the authors provide four theorems with proofs that analyze the bounded representation discrepancy, global position encoding distinctness, noise robustness of boundary-aware convolution, and generalization guarantees for tree perturbation. Experimentally, results on four real-world datasets and four synthetic datasets show that GPEN performs well on subgraph representation learning tasks. The ablation study (Table 4) demonstrates the contribution of each component, validating the complementary nature of the three modules: Global Position Encoding (GPE), Boundary-Aware Convolution (BWC), and Optional Tree Perturbation (OTP).

**Essential References Not Discussed:**

The paper discusses most of the relevant important references. The authors' review of related work covers the progression from fundamental work on Graph Neural Networks to advances in subgraph representation learning methods.

**Experimental Designs Or Analyses:**

The experimental design is reasonably sound. The authors adopt the same datasets and data splits as the baselines, facilitating fair comparison. The experiments cover multiple aspects: performance comparison on real-world datasets (Table 1), controlled experiments on synthetic datasets (Table 3), ablation studies (Table 4), and hyperparameter analysis. The authors explore the impact of different tree construction algorithms (Table 6), comparing Breadth-First Search Tree, Depth-First Search Tree, Minimum Spanning Tree, and Maximum Spanning Tree algorithms.

**Methods And Evaluation Criteria:**

The proposed methods and evaluation criteria are appropriate for the research problem. The authors use the same datasets and data splits as previous work, ensuring comparability of results. The evaluation uses micro-F1 scores as metrics, which is a common measure for subgraph classification tasks. Experiments are conducted on various datasets with different characteristics, including molecular biology, clinical diagnostics, and user profiling datasets from real-world scenarios, as well as synthetic datasets designed to test the ability to recognize different structural features. Notably, the authors explore the impact of different tree construction algorithms (Table 6), demonstrating the superiority of the Maximum Spanning Tree algorithm, which enhances the credibility of their method selection.

**Other Comments Or Suggestions:**

Some figures in the paper could be improved, such as the y-axis labels and line colors in Figure 4. Consider adding grid lines or using different line types to distinguish different data series, which would help readers better interpret the results.

**Other Strengths And Weaknesses:**

Strengths:
1. The global position encoding proposed in GPEN is a novel approach that systematically captures multi-hop relationships between nodes in a graph.
2. The paper provides comprehensive and rigorous theoretical analysis, proving the effectiveness and stability of the method.
3. Experiments are conducted on multiple datasets, validating various aspects of the method through ablation studies and hyperparameter analysis.
4. The paper mentions using COO-formatted sparse adjacency matrices to calculate difference vectors of node representations, demonstrating consideration for memory efficiency.

Weaknesses:
1. Some figures' labels and color choices could be clearer, particularly the visualizations in Figure 4.
2. The paper discusses the impact of threshold c but lacks methods for automatically determining the optimal threshold.

**Questions For Authors:**

Among the different tree construction algorithms, the Maximum Spanning Tree performs relatively well. Is this advantage consistent across all types of graph structures, or is it more significant for certain types of graphs (such as sparse or dense graphs)?

**Relation To Broader Scientific Literature:**

The paper has connections to the scientific literature. In Section 6, the authors review related work, including the development of Graph Neural Networks and advances in subgraph representation learning. The paper discusses existing methods such as SubGNN, GLASS, SSNP, and S2N, and explains how GPEN attempts to address some of the challenges faced by these methods from the perspectives of global structural information and selective information integration. This literature connection helps to understand the research background and contributions of GPEN.

**Theoretical Claims:**

The theoretical claims in the paper are supported by sound mathematical proofs. Section 4 provides four core theorems: Theorem 4.1 proves that the discrepancy between GPEN representations and standard GNN representations is bounded; Theorem 4.2 establishes that global position encoding effectively distinguishes nodes with different connectivity patterns; Theorem 4.3 analyzes the noise robustness of boundary-aware convolution, proving it achieves higher signal-to-noise ratio compared to standard GNN aggregation; and Theorem 4.4 provides generalization bounds for the tree perturbation technique. The proofs are elaborated in Appendix A.1 with rigorous derivations using appropriate mathematical tools such as Lipschitz continuity, Perron-Frobenius theorem, and PAC-Bayes framework.

---

> ### Author Rebuttal · Authors · 2025-03-31
>
> We thank the reviewer for their detailed comments. We appreciate that they found our "theoretical claims in the paper are supported by sound mathematical proofs" and acknowledged that our "experimental design is reasonably sound." Below, we have responded to the weaknesses raised by the reviewer:
>
>
> **Weakness 1 & Other Comments/Suggestions: Improving Figure Clarity**
>
>
> Thank you for your suggestion regarding the color coding in the captions of tables. For the revised manuscript, we will:
> - Enhance the clarity of y-axis labels and line colors in Figure 4.
> - Add grid lines to facilitate easier interpretation of results in Figure 4.
>
> These visual improvements will make our results more accessible and interpretable for readers.
>
>
>
> **Weakness 2: Methods for Automatic Threshold Determination**
>
>
> We agree with the reviewer that such methods can be very insightful. However, we would like to note that since the tree perturbation module is optional (as not all datasets suffer from insufficient samples), and as reviewer FTVY13 noted, we don't generate too many samples, making manual parameter tuning sufficient for most applications.
>
>
> In Figure 3 of our manuscript, we presented a comprehensive analysis of how different threshold values $c$ affect model performance. For enhanced clarity, we have reformatted these experimental results in tabular form below:
>
>
> | Threshold c value | 2 | 4 | 6 | 8 |
> |---------|---------|---------|---------|---------|
> | ppi_bp | 0.611 | 0.622 | 0.644 | 0.633 |
> | hpo_metab | 0.638 | 0.610 | 0.603 | 0.598 |
> | hpo_neuro | 0.671 | 0.691 | 0.681 | 0.681 |
> | em_user | 0.876 | 0.903 | 0.912 | 0.902 |
>
>
> This table reveals that while threshold selection influences performance, the variation is generally moderate. GPEN demonstrates relatively stable performance across a reasonable range of threshold values (c=4 to c=8), suggesting it is not overly sensitive to the exact choice of c. This stability reduces the need for precise automatic tuning. For the hpo_neuro dataset, we observe identical performance at thresholds 6 and 8 because there are no remaining connected components larger than these thresholds, which inherently limits the number of samples our method generates. This natural upper bound on generated samples further reduces the necessity for complex automatic threshold determination methods. In addition, the tree perturbation module is optional and primarily benefits datasets with insufficient samples. For datasets with abundant samples, this module may not be necessary.
>
> **Question: Maximum Spanning Tree Performance Across Graph Types**
>
>
> Thank you for this insightful question about the consistency of Maximum Spanning Tree (MaxST) performance across different graph structures.
> In Appendix A.2.4, we conducted a comprehensive analysis of different tree construction algorithms and their impact on GPEN's performance. As shown in Table 6 of our appendix, we evaluated four representative tree construction methods: Breadth-First Search Tree (BFS), Depth-First Search Tree (DFS), Minimum Spanning Tree (MST), and Maximum Spanning Tree (MaxST). Our analysis shows that while MaxST consistently outperforms other tree construction algorithms across all datasets, its advantage is indeed more pronounced in certain graph types.
>
> The experimental results show that the Maximum Spanning Tree algorithm consistently achieves superior performance across all datasets. This is primarily because MaxST effectively preserves critical structural information during the tree construction process by utilizing node importance scores derived from PageRank. The weighting scheme ensures that connections between high-importance nodes are prioritized in the resulting tree structure, effectively capturing the backbone of the graph's hierarchical organization. As reviewer FTVY13 noted, this "effectively clarifies the practical significance of utilizing a hierarchical tree in the study."
>
> The advantage of Maximum Spanning Tree is particularly pronounced in dense graphs with higher average node degrees. For instance, in dense networks such as hpo-metab and hpo-neuro, MaxST achieves significantly better results (0.638 ± 0.009 and 0.691 ± 0.006 respectively) compared to BFS (0.515 ± 0.031 and 0.606 ± 0.037) and DFS (0.494 ± 0.030 and 0.605 ± 0.034). This performance advantage stems from MaxST's ability to identify and preserve important pathways in complex network topologies, resulting in more meaningful global position encodings that better capture the structural relationships between distant nodes.

---

### Official Review · Reviewer_FTVY · 2025-03-13

**Overall Recommendation:** 3

**Summary:**

The paper introduces a method called GPEN for Subgraph Representation Learning. It proposes the construction of a hierarchical tree to compute the Global Position Encoding (GPE) and introduces Boundary-aware Convolution (BWC) and tree-based Optional Tree Perturbation (OTP). These strategies aim to address two major challenges in graph representation learning: capturing structural relationships between distant nodes and preventing excessive aggregation of global structural information.

**Claims And Evidence:**

The paper provides detailed theoretical analysis and validation for the GPE, BWC, and OTP in GPEN.

**Essential References Not Discussed:**

N/A

**Experimental Designs Or Analyses:**

The paper demonstrates through experiments that the standard deviation of GPEN is significantly lower than other methods, indicating better robustness. However, several conclusions from the experiments are not significant, as detailed in comment W4.

**Methods And Evaluation Criteria:**

Experiments show that GPEN has better average performance and lower standard deviation. However, the experimental results are not significant, as detailed in [W4].

**Other Comments Or Suggestions:**

In section 3.1.2, the global positional encoding groups nodes according to their depth in the tree, then encodes them directly into a 01 vector. It loses the information of its value. Especially considering that the edge weights and trees here are artificially calculated and constructed, the shortest path length on the tree may not represent real and accurate global information. Would it be better to use an encoding that retains the depth value of the nodes, such as the sin/cos function encoding designed for sequences in the classic "Attention is all you need"?

**Other Strengths And Weaknesses:**

Strengths:

The paper provides detailed theoretical analysis and validation for each module.

Weaknesses:

[W1] In section 3.1.1, as the transition matrix of PageRank, why $M$ is $\mathbf M_{ij}=\frac{1}{d_i}$ instead of $\frac{1}{d_j}$? Also, what is the initial value of $R$ in equation 3?

[W2] Section 3.1.3 mentions that "The insufficient number of subgraphs can affect the model’s stability." However, the tree perturbation module generates at most one new sample for each subgraph. Therefore, the total number of subgraphs is at most twice the original number. Is this sufficient to address the issue of insufficient subgraphs?

[W3] In section 3.2, the notion in equation 12 is unclear:
* What is $\mathbf A$? It seems to be undefined in the paper.
* What does the subscript $n$ in $\mathbf H_n^{(l-1)}$ stand for? Is it a different matrix from $\mathbf H^{(l-1)}$?
* Is $\mathbf W^{(l-1)}$ the edge weight of the weighted graph in section 3.1.1? If so, what does the superscript $(l-1)$ represent? The edge weight in section 3.1.1 seems to be a constant independent of the layer.

[W4] In Table 1, the $p$-values comparing GPEN with S2N on all datasets are greater than 0.05, indicating non-significant differences. The same applies to the experiments in Table 3 and the ablation study in Table 4. The advantage of GPEN seems to lie only in its more stable performance and smaller variance.

**Questions For Authors:**

See above.

**Relation To Broader Scientific Literature:**

The paper identifies two major challenges in existing Subgraph Representation Learning methods: capturing structural relationships between distant nodes and preventing excessive aggregation of global structural information. It proposes to construct a hierarchical tree and designs the GPE, BWC, and OTP modules to address these challenges.

**Theoretical Claims:**

The paper proposes several theorems as theoretical support for the effectiveness of the modules in GPEN, and provides comprehensive proofs in the appendix.

---

> ### Author Rebuttal · Authors · 2025-03-31
>
> We sincerely appreciate your positive feedback and constructive remarks on our paper. Below, we provide a detailed response to your questions and comments.
>
> **[W1 and W3]**
>
> We thank the reviewer for pointing out these notation issues throughout the paper. To clarify:
> - $M_{ij}$ : Thank you for bringing this to our attention. You are correct; $M_{ij} = \frac{1}{d_j}$ if there is an edge between nodes $i$ and $j$, where $d_j$ is the out-degree of node $j$. The incorrect notation in the paper was a typographical error.
> - $R$: The initial value of $R$ in equation 3 is a uniform distribution where each element equals $\frac{1}{|V|}$, with $|V|$ being the number of nodes in the graph.
> - $\mathbf{A}$: This represents the adjacency matrix of the graph, where $A_{ij} = 1$ if there is an edge between nodes $i$ and $j$, and $A_{ij} = 0$ otherwise.
> - $\mathbf{H}_n^{(l-1)}$: The subscript $n$ was a typographical error. It should be $\mathbf{H}^{(l-1)}$, representing the node embeddings from layer $l-1$.
> - $\mathbf{W}^{(l-1)}$: This refers to the trainable parameter matrix in the graph convolution operation, not the edge weights $w_{ij}$ mentioned in section 3.1.1. The superscript $(l-1)$ indicates that this parameter matrix is associated with the transformation from layer $l-1$ to layer $l$.
>
> We will correct these notations in the revised manuscript and use more distinctive symbols to avoid confusion.
>
> **[W2]**
>
> Our experimental results show that tree perturbation is effective with modest sample increases, though excessive samples may introduce noise. As shown in Figure 3 in our paper, we conducted detailed experiments on the impact of different threshold values c on model performance. Due to space limitations, we present the numerical results in our response to Reviewer 2fVD under "Weakness 2: Methods for Automatic Threshold Determination." These results clearly identify that even a relatively small number of additional samples can significantly improve the model's performance, while excessive samples may actually harm performance.
>
> We agree that this is an important consideration. However, our primary goal is to explore the potential of tree structures in subgraph representation learning. While our results show promising improvements, we do not claim this approach "completely solves the issue of insufficient subgraphs" in our paper. We recognize that developing high-quality data augmentation methods requires consideration of many factors, which is beyond the scope of our current work.
>
> **[W4]**
>
> We thank the reviewer for the detailed statistical analysis. However, we respectfully argue that a holistic evaluation reveals significant advantages for GPEN.
>
> Crucially, GPEN shows superior performance over S2N on most datasets, particularly on synthetic datasets specifically designed to evaluate distinct structural learning capabilities. On these challenging tasks, GPEN outperforms all baseline methods, showcasing its robust structural understanding. In stark contrast, S2N not only fails to match GPEN but actually exhibits performance degradation compared to other established baselines. For example, on cut-ratio, S2N scores 0.892 vs. GLASS's 0.935 vs. GPEN's 0.936, and on coreness, S2N scores 0.726 vs. GLASS's 0.840 vs. GPEN's 0.876.
>
> While S2N achieves comparable mean results on a few datasets, it does so with markedly high standard deviations compared to other baselines (averaging a 43.2% increase over SubGNN and a 127.7% increase over GLASS). GPEN, conversely, achieves its strong performance with significantly lower standard deviations across the board (averaging a 59.8% reduction compared to SubGNN and 37.5% compared to GLASS).
>
> **[S1]**
>
> Thank you for this insightful suggestion regarding alternative encoding methods for the global positional information. Our choice of one-hot encoding for tree depths was based on several practical considerations:
>
> First, the inherent message-passing mechanisms in GNNs enable the model to naturally learn relationships between different depth levels during convolution operations, making explicit continuous encoding less critical in this context. Second, node depth primarily functions as a categorical feature that distinguishes nodes. The one-hot representation provides clear separation between these structural groups. Third, unlike transformer models that should generalize across unseen positions, our model operates on specific graph structures where the positional relationships are fixed, reducing the need for the interpolation capabilities that make sin/cos function encodings valuable in sequence modeling.
>
> We conducted additional experiments comparing one-hot encoding with sin/cos function encodings, but due to space limitations, we apologize that we couldn't present these results. Consistent with our reasoning, these experiments showed no significant performance improvements, as both encoding methods capture the same fundamental structural information.

---

### Decision · Program_Chairs · 2025-05-01

**Decision:**

Accept (poster)

**Comment:**

This paper introduces GPEN (Global Position Encoding Network), a novel method aimed at enhancing subgraph representation learning (learning the representation of a subgraph in a graph) by effectively capturing global structural information through hierarchical tree-based position encoding and a boundary-aware convolution module. The reviewers generally appreciate the clear motivation, rigorous theoretical analysis, and thorough empirical validation across diverse datasets. Two reviewers recommended acceptance (score 4), while two give weak accept (score 3). Given the important problem of subgraph representation learning, high-quality theoretical contributions, comprehensive empirical validation, clarity of writing, and thorough responses addressing reviewer concerns, the consensus is to recommend acceptance of the paper, subject to the inclusion of the promised clarifications and improvements in the final manuscript.